# English Adverbial and Determiner Negation: A Problematic Area for Arabic Translators

**Mohammed Farghal**

Department of English, Kuwait University, Jamal Abdul Nasser St, Kuwait City 13060, Kuwait; m_farghal@hotmail.com

**Abstract:** Negation hardly comes up as an issue in English–Arabic translation studies. The general assumption is that the translation of English negation into Arabic poses no serious problems to the translator. While this is generally true when it comes to rendering negation marked by generic negative particles/affixes (*John is not happy* and *John is unhappy*, respectively) and even lexical and rhetorical implicit negation (*John denied having cheated on the test* and *Can a person like John make such a mistake?*), the present paper aims to show that the appropriate textualization into Arabic of English adverbial and determiner negation (e.g., by the adverbials *too* and *hardly*, and the determiners *little* and *few*) can be a problematic area for Arabic translators. The textual data (270 examples) is extracted from several published translations (belonging to literary, popular science/journalistic, and economic discourse), in an attempt to show what strategies translators follow when encountering such negation and how successful they are. While the findings provide solid evidence for the serious mishaps (about 42% of the renderings involve one kind of problem or another) that Arabic translators experience in this area, the critical discussion unravels several textual strategies that can capture the subtleties inherent in adverbial/determiner negation. It is hoped that the investigation of this subtle, neglected area in English–Arabic translation studies offers significant insights for both student and professional translators.

**Keywords:** adverbial negation; determiner negation; English; Arabic; translation

## 1. Introduction

There have been several studies which deal, among other things, with the linguistics of negation in Arabic, where negation is divided into explicit and implicit negation (Anees 1975; Al-Makhzumi 2016; Alsalem 2012; Muslah 2015). While explicit Arabic negation employs negative particles such as *lam* [did not], *laa* [do not], and *lan* [will not], implicit Arabic negation uses grammatical devices such as interrogatives and conditionals. However, there are only very few studies that have dealt with Arabic and English negation from a translational perspective (e.g., Dendane and Dendane 2012; Al-Ghazalli 2013). In particular, there are no studies, to my knowledge, that raise the question of English non-generic explicit negation by *too* and *-ly* adverbs and *little* and *few* determiners, which is supposed to be a problematic area because Arabic lacks this type of explicit negation.

The present study, therefore, aims to fill in this gap in English–Arabic translation studies by addressing itself to the translation of non-generic adverbial and determiner negation, as an area of contrast between English and Arabic that may cause serious problems. While English uses both explicit generic negation (negation by *not* and similar negative particles) and explicit non-generic adverbial/determiner negation, Arabic lacks the latter type as it exclusively employs explicit generic negation by negative particles, such as *laa*, *laysa*, *lam*, *maa*, *lan*, etc., which are all tense sensitive (*laa* and *laysa* in present, *lam* and *maa* in past (with *lam* [followed by imperfective verb form being used much more commonly in Modern Standard Arabic than *maa* (followed by perfective verb form)], and *lan*

in future). It seems necessary, in such case, to examine English non-generic adverbial/determiner negation from a translational perspective in order to explore the translation strategies that translators resort to, when modulating this type of negation, in an attempt to capture its meaning and pragmatics. Such investigation is expected to have theoretical as well as practical implications for people working in English–Arabic translation studies, and may also prompt future investigations between English and other languages that lack this type of negation.

The study is structured as follows. Section 2 reviews the related literature from both a linguistic and a translational perspective. Section 3 presents the research questions. Section 4 describes the textual materials used. Section 5 offers a detailed analysis of the data. Section 6 discusses the findings. Finally, Section 7 concludes the study.

## 2. Review of Literature

Negation, whose universality is unanimously confirmed in the existing literature on human language (Dahl 1979; Payne 1985; Horn and Kato 2000; Horn 2001, among others), is generally considered an operator that reverses the truth value of a proposition. It is a unique property of human language: "Negative utterances are a core feature of every system of human communication and of no system of animal communication" (Horn and Kato 2000, p. 1). Linguists (Klima 1964; Clark 1976; Horn 2001, among others) usually divide negation into two types: explicit negation and implicit negation. On the one hand, explicit negation employs explicit negative particles such as *not* (as in *John did not go shopping*), negative affixes such as *-il* in *This act is illegal*, or other negative adverbs, for example, *hardly* (as in *John hardly knows anything about mathematics*) or determiners, for example, *few* (as in *John has only few friends*). On the other hand, implicit negation is implied semantically (as in *John **prevented** his daughter from joining the club*, which semantically entails *John **did not allow** his daughter to join the club*) or implicated pragmatically (as in the rhetorical question *Should we keep silent after all these heinous crimes*, which conversationally implies *We should not keep silent after all these heinous crimes*).

Similarly, Arabic negation is divided into explicit and implicit negation. While English verbal negation is uniform in nature, as it only employs the negative particle *not* in such negation (viz. *John does not go to school*, *John did not go to school*, and *John will not go to school*), its Arabic explicit counterpart is highly diversified (for details about Arabic negation, see Anees 1975; Al-Makhzumi 2016; Alsalem 2012; Muslah 2015), viz. لا يذهب جون إلى المدرسة *laa yaðhabu joon 'ilaa al-madrasati* [not (present) go John to the school], لم يذهب جون إلى المدرسة *lam yaðhab joon 'ilaa al-madrasati* [not (past) go John to the school], لن يذهب جون إلى المدرسة lan *yaðhaba joon 'ilaa al-madrasati* [not (future) go John to the school]. Al-Makhzumi (2016, p. 265), for example, defines negation as "a linguistic category which is opposed to affirmation and intended to disprove or deny the truth value of a proposition". Explicit Arabic negation employs negative particles such as *lam* and *maa* [did not] (e.g., لم يكتب/ ما كتب سالم الرسالة *lam yaktub/maa kataba Saalimun 'ar-risaalata* [Negative particle (NEG) write/NEG wrote Salim the-letter] "Salim did not write the letter", *laa/laysa* [do not/is not] (e.g., لا يكتب سالم رسائل *laa yaktubu Saalimun rasaa'ila* [NEG write Salim letters] "Salim does not write letters" and ليس الكتاب جيدا *laysa-l-kitaabu jayyidan* [NEG-the-book good] "The book is not good", and *lan* [will not] (e.g., لن يكتب سالم رسالة *lan yaktuba Saalimun risaalatan*) "Salim will not write a letter". By contrast, implicit Arabic negation uses grammatical devices such as interrogatives (e.g., هل يتساوى الخير والشر *hal yatasaawa-l-xayru wa-š-šarru*? [Question word (Q) equal-the-good and-the-evil] "Are the good and evil equal?" and conditionals (e.g., لو كنت موجوداً لأخبرته *law kunto mawjuudan la-'xbartu-hu* [if was(I) present so-told-him(I)); both examples involve implicit negation, viz. the answer to the question must be in the negative (No, they are not equal), and the latter statement implies that the speaker had not been there. Nevertheless, this fact does not cause any serious problems for the translator into Arabic because the one-to-many correspondence between the negative particles is obvious, being tense oriented in Arabic.

Notably, while Arabic translation correspondents are usually accessible for generic negation, e.g., *John does not adhere to punctuality*/لا يلتزم جون بالمواعيد *laa yaltazimu joon bi-l-mawaa'iidi* [not (present) adhere John to-the-punctuality], affixal negation, for example, *John is unhappy*/جون غير سعيد *joon ɣayru sa'iidin* [joon not happy], and implicit negation (Arabic being as highly lexicalized as English), for example, *John declined the offer*/رفض جون العرض *rafaḍa joon al-'arḍa* [declined john the-offer], which semantically entails *John did not accept the offer* and لم يقبل جون العرض *lam yaqbal joon al-'arḍa* [not (past) accept John the-offer], respectively. However, the translator is required to modulate ([Vinay and Darbelnet 1995](#)) or look for translation equivalents ([Koller 1979](#)) in the case of adverbial/determiner negation by usually retrieving negation (whether explicit or implicit) in Arabic. For example, the adverbial negation by *too* in *John was too ambitious* needs to be modulated by recovering generic negation in Arabic as in لم يكن جون واقعياً في طموحاته *lam yakun joon waaqi'iyyan fii ṭumuuḥaati-hi* [not (past) be John realistic in ambitions-his] or implicit negation as in تجاوز جون الحدود في طموحاته *tajaawaza joon al-ḥuduuda fii ṭumuuḥaati-hi* [exceeded John the-limits in ambitions-his]. Similarly, the determiner negation in *John has little interest in politics* needs to be modulated into generic negation as in ليس لدى جون إلا القليل من الاهتمام في السياسة *laysa ladaa joon 'illa-l-qaliilu min al-'ihtimaami fis-siyaasati* [not (present) with John except little from interest in the-politics] or لا يهتم جون إلا قليلاً في السياسة *laa yahtammu joon 'illaa qaliilan fis-siyaasati* [not (present) be interested John except little in the-politics], or implicit negation as in ينأى جون بنفسه عن الاهتمام في السياسة *yan'aa joon bi-nafsihi 'an-il-'ihtimaami fis-siyaasati* [alienates John with-himself from the-interest in the-politics].

In terms of translation, there are only very few studies on the translation of negation. [Apostolatu and Apostolatu](#) ([2012](#)) deal with literary translation of English negation into Romanian. They show that some negative markers are sometimes unjustifiably omitted, which is usually caused by the differences between the two languages involving negative polarity, scope of negation, and double negation. [Dendane and Dendane](#) ([2012](#)) refer to the one-to-many correspondence between the English particle *not* and the many counterparts in both standard and vernacular Arabic, which causes serious problems to machine rather than human translation. [Li](#) ([2017](#)) points out the difficulty Chinese English foreign language (EFL) learners face when expressing adverbial negation by *too* due to its Chinese counterpart, which functions as an intensifier. Hence, Chinese learners often erroneously employ the negative adverb *too* instead of the intensifier "very" or "so" (e.g., "The party was too good" may be used to mean "The party was very/so good").

A similar mishap may occur in English-into-Arabic translation. [Farghal and Almanna](#) ([2015a](#), p. 27) briefly examine negation while discussing syntactic features in translation. Whereas they state that English generic negation by *not* is not problematic when rendering it into Arabic, despite the existing one-to-many correspondences, it is argued that the negation embedded in *too* can pose a challenge because it requires a translation strategy that recovers negation in Arabic, whether explicitly or implicitly. To demonstrate this point, they give the following example from a published translation in which the negation is missed by replacing the negative adverb *too* with the Arabic intensifiers *jiddan* [very] (a list of Arabic phonetic symbols (mainly International Phonetic Alphabet (IPA)) is provided in Appendix [A](#)):

1.  I think you've been too busy to notice where I have been.
    'aðunnu 'anna-ka kunta maš'uulan　　jiddan li-tulaaḥiða　　'ayna 'anaa
    think (I) that-you be-you busy　　　　very to-notice　　　　where I
    "I think you were too busy to notice where I am".

    أظن أنك كنت مشغولاً جداً لتلاحظ أين أنا.

[Al-Ghazalli](#) ([2013](#)) discusses the translation of Arabic implicit negation where he unjustifiably argues for unpacking Quranic implicit negation in rhetorical questions. According to his analysis, the

Quranic verse هل يستوي الأعمى والبصير *hal yastawi-l-ʾaʿmaa wa-l-baṣiir* [Q equal the-blind and-the-sighted] is erroneously rendered as a generic rather than a rhetorical question by Quran translators as in Yousef Ali's *Can the blind be held equal to the seeing* (p. 135) and M. Pickthal's *Are the blind and the seer equal* (p. 133). Therefore, he claims, implicit negation should be made explicit as in *Are the blind and the one who sees equal? Definitely, this is not true* (p. 139). Needless to say here, that mainstream translation theorists (Nida 1964; Catford 1965; Newmark 1988; Baker 1992; Hatim 1997; Dickins et al. 2002; Pym 2010; Munday 2012; Farghal 2012; Farghal and Almanna 2015b; Farghal et al. 2015, among others) emphasize the translator's ability to call up textual/functional material in the target language (TL) that effectively relays its counterpart in the source language (SL). One should note that, textually as well as functionally, Arabic rhetorical questions readily translate into English rhetorical questions (Ali's and Pickthal's above), thus remaining within the scope of implicit rather than explicit negation.

## 3. Research Questions

Due to the fact that English adverbial and determiner negation has no formal translational equivalent in Arabic and the lack of studies in this area, the purpose of this study is to explore the translation strategies that translators employ when rendering such negation into Modern Standard Arabic and examine how successful these strategies are. In particular, the study aims to check whether Arabic translators retrieve explicit negation (or alternatively use implicit negation) when modulating adverbial and determiner negation in an attempt to capture the pragmatics of this type of negation. Specifically, the following research questions are addressed in this paper:

1. How do Arabic translators tackle *too* adverbial negation in terms of translation strategies?
2. How do Arabic translators render English *-ly* adverbial negation in terms of translation strategies?
3. What translation strategies do Arabic translators follow in rendering *little* determiner negation?
4. Finally, are Arabic translators sensitive to *few* determiner negation?

## 4. Textual Materials

This is an empirical study based on the extraction of ample textual data involving English adverbial/determiner negation along with their target Arabic counterparts from existing works and their translations. The textual data (270 examples) features two types of markers of adverbial negation (*too* and *hardly/scarcely/barely*) and two markers of determiner negation (*little* and *few*). The study provides both a quantitative and a qualitative analysis of the data. It should be noted that the qualitative discussion almost exclusively focuses on the rendering of the items under study which belong to adverbial and determiner negation, apart from the general quality of the translation, which is not within the scope of this study.

The sources of the textual data include five series of Harry Potter (HP, henceforth) by J. K. Rowling, namely *Harry Potter and the Philosopher's Stone* (1; Rowling 1997), *Harry Potter and the Prisoner of Azkaban* (2; Rowling 1999), *Harry Potter and the Chamber of Secrets* (3; Rowling 1998), *Harry Potter and the Order of Phoenix* (4; Rowling 2003), and *Harry and the Half-blood Prince* (5; Rowling 2005). The first and the third series are translated by Ragaa Abudullah. The second series is translated by Hasan Ahmed Mohammed, the fourth by Idaarit Al-Nashr and the fifth by Abd Al-Wahab Aloob (see references for complete information). The textual data sources also include *The Blue Flower* (BF) by Fitzgerald (1995) (translated by Ali Suleiman), *The Fault in our Stars* (FS) by Green (2012) (translated by Baseel Intwan), *The Help* (TH) by Stockett (2009) (translated by Hassan Al-Bustani), *The Future: Six Drives of Global Change* (GC) by Gore (2013) (translated by Adnan Gergeos), and *The Making of Economic Society* (ES) by Heilbroner (1962) (translated by Rashid Al-Barrawi).

The choice of the textual data is motivated by the different genres it belongs to, viz. literary, popular science/journalistic, and economic, as well as the different translators involved in translating it. The aim is to investigate a representative sample of textual material in terms of genre and translators

in order to come up with generalizations about the translation of adverbial and determiner negation across several genres and translators.

## 5. Analysis

The analysis examined the translation strategies employed by the translators in rendering adverbial and determiner negation in terms of frequency and percentage, which gave a clear picture about their utility when encountering such negation. First, Section 5.1 presented the markers of adverbial and determiner negation and their distribution in the English corpus. Second, Section 5.2 examined the translation strategies employed in rendering *too* adverbial negation, using both explicit negation (Section 5.2.1) and implicit negation (Section 5.2.2). Third, Section 5.3 looked at how the translators dealt with -ly adverbial negation in terms of translation strategy. Fourth, Section 5.4.1 investigated how the translators had tackled *little* determiner negation and the translation strategies adopted. Finally, Section 5.4.2 considered *few* determiner negation and how sensitive the translators had been to this type of negation.

### 5.1. Adverbial and Determiner Negative Markers

In terms of the type of negative marker, the English corpus is distributed as shown in Table 1 below.

**Table 1.** Distribution of English negative markers in the corpus.

|   | Marker | Frequency | Percentage |
|---|--------|-----------|------------|
| 1 | Too | 100 | 37% |
| 2 | Hardly | 34 | 12.60% |
| 3 | Scarcely | 19 | 7% |
| 4 | Barely | 11 | 4% |
| 5 | Little | 86 | 31.85% |
| 6 | Few | 20 | 7.40% |
|   | Total | 270 | 100% |

Table 1 shows that adverbial negation by *too* is the most frequent in the corpus (37%) followed by determiner negation by *little* (31.85%). Third comes adverbial -ly negation including *hardly*, *scarcely,* and *barely* which together account for (23%). Within -ly negation, *hardly* emerges as the most frequent (12.60%), followed by *scarcely* (7%), then *barely* (4%). The least frequent in the data is determiner negation by *few,* which only accounts for (7.40%). These percentages may only give us a preliminary picture about the frequency of adverbial/determiner negation in English discourse. To affirm such frequencies, a large scale quantitative and qualitative corpus linguistics investigation needs to be carried out, which is far beyond the scope of the present study.

### 5.2. Translating Adverbial Negation by Too

### 5.2.1. Explicit Negation

Table 2 below presents the frequency and percentage of employing translation strategies in rendering *too* adverbial negation.

**Table 2.** Frequency and percentage of explicit negation strategies in rendering *too* negation.

| Translation Strategy | Frequency | Percentage |
|----------------------|-----------|------------|
| a. Unpacking by coordination | 17 | 36.17% |
| b. Nominalization in simple/complex structures | 15 | 31.91% |
| c. Indicating degree of attribute | 11 | 23.40% |
| d. Mistranslations/under-translations | 4 | 8.51% |

Table 2 above shows that there are three main translation strategies adopted by Arabic translators when opting to render English *too* negation by explicit Arabic negation: (1) Unpacking by coordination, (2) nominalization in simple or complex structures, and (3) indicating degree of attribute.

Firstly, unpacking *too* negation by a coordinate Arabic structure featuring explicit negation is the most frequent translation strategy at 36.17% (17 cases) for rendering *too* negation. It proves to be a workable strategy, as can be observed in the following example:

2.  Harry was too deeply asleep to hear her. (HP/4)
    haarii        kaana        ɣaariqan       fii nawmi-hi fa-lam yasmaʿa-haa
    Harry         was          sinking        in sleep-his so-not hear-her
    "Harry was deeply asleep, so he didn't hear her"

    هاري كان غارقاً في نومه فلم يسمعها.

It is possible, also, to relay *too* negation in such cases by maintaining the English complex structure (second strategy below), as is shown in the rephrasing of (2) below, with a shift of focus in the ordering of the two propositions:

3.  lam yasmaʿ-haa        haarii li'anna-hu        kaana ɣaariqan fii nawmi-hi
    not hear-her           Harry because-he         was sinking in sleep-his
    "Harry didn't hear her because he was deeply asleep"

    لم يسمعها هاري لأنه كان غارقاً في نومه.

Secondly, nominalization in simple or complex Arabic structures comes second at 31.91% (15 cases). The translators have mostly succeeded in capturing *too* negation using this strategy, as can be noted in the examples below:

4.  I was too late to save the girl. (HP/3)
    lam 'astatiʿ          'inqaað-l-fataati fi-l-waqti-l-munaasib
    not be able           saving-the-girl-in-the-time-suitable
    "I couldn't save the girl at the right time".

    لم أستطع إنقاذ الفتاة في الوقت المناسب.

5.  ... but his Patronus was too feeble to drive the dementor away. (HP/2)
    wa-laakinna          taʿwiiðata-hu        lam takun        bi-l-quwwati-l-kaafiyati
    and-but              Patronus-his         not be           with-the-strength-the-enough
    li-'ibʿaadi-l-ḥaaris
    to-driving away-the-dementor
    " ... but his Patronus wasn't strong enough to drive the dementor away".

    ولكن تعويذته لم تكن بالقوة الكافية لإبعاد الحارس.

Thirdly, capturing *too* negation by indicating the degree of the attribute in question accounts for 23.40% (12 cases) of the examples in this category. Semantically, it corresponds to awkwardly rephrasing *too* negation by using the phrase *to the extent that* with negation by *not* in English, viz. *John was too short to touch the ceiling* may awkwardly be rephrased as *John was short to the extent that he couldn't touch the ceiling*. In Arabic, this strategy proves very useful for rendering *too* negation. The following example is illustrative:

6.  Professor Trelawney seemed too tipsy to have recognized Harry. (HP/5)
    badat al-'ustaaðatu triiloonii maxmuuratun li-darajati 'anna-ha lam taʿrif haarii
    seemed the-professor Trelawney drunk to-extent that-she not know Harry
    "Professor Trelawney was drunk to the extent that she didn't recognize Harry".

    بدت الأستاذة تريلوني مخمورة لدرجة أنها لم تعرف هاري.

Finally, let us look at one example where the translator has failed to capture the subtlety of *too* negation by reducing it to generic Arabic negation, viz. " . . . but Harry was too used to this to care" is rendered into ولكن هاري لم يهتم بهذا *wa-laakina haari lam yahtam bi-haaðaa* [and-but Harry NEG care in-this] "but Harry did care about this". As can be seen, the translator has managed to recover explicit Arabic negation but failed to capture the meaning of *too* negation. To do this, one may suggest ولكن هاري لم يهتم لتعوّده على هذا *wa-laakina haari lam yahtam li-taʿawwudi-hi ʿalaa haaðaa* [and-but Harry NEG care because-being used to-him on this] " . . . but Harry did not care because he was used to this".

### 5.2.2. Implicit Negation

Table 3 below displays the frequency and percentage of using implicit negation translation strategies when rendering *too* adverbial negation.

**Table 3.** Frequency and percentage of implicit negation strategies in rendering *too* negation.

| Translation Strategy | Frequency | Percentage |
| --- | --- | --- |
| a. Using comparative form | 16 | 28.80% |
| b. Using negative verbs | 11 | 20.75% |
| c. Indicating degree of attribute | 10 | 18.87% |
| d. Mixed bag (mistranslations/under-translations) | 16 | 30.19% |

Apart from the mixed bag of erroneous translations/under-translations (16 instances, 30.19%), Table 3 indicates that there are three main implicit negation strategies for capturing the meaning of *too* negation: (1) Employing the comparative form, (2) employing negative verbs, and (3) indicating degree of attribute.

The translation strategy using the Arabic comparative form (16 instances, 28–30%) emerges as very useful for handling *too* negation. The comparative Arabic forms *'afʿal min* [more of an attribute than] and *'akθar maṣdar* (verbal noun) *min* [more of verbal noun than] here capture the nuance that the force of X's attribute goes beyond the capability of Act Y, for example, *'aðkaa min 'an yuxdaʿ* [cleverer than that (he) be deceived] and *'akθara ðakaa'an min 'an yuxdaʿ* [(has) more cleverness than that (he) be deceived] which both idiomatically translate into *Ali is too clever to deceive*. Following are two illustrative examples from the corpus:

7. . . . because it was too long to memorize. (FS)
   . . . li'anna-ha   'aṭwalu min 'an   tuḥfaǧa   ɣayban
   . . . because-it   longer from that   be learned   by heart
   " . . . because it was longer than it could be memorized".

<div dir="rtl">

. . . لأَنها أطول من تُحفظ غيباً.

</div>

8. My way of life here is pitched too high for his young head. (BF)
   'inna ḥayaatii   hunaa   'akθara   ṣaxaban min 'an yaḥtammalu-ha ra'sa-hu-l-yaafiʿ
   verily life-my   here   More   noisy from that tolerated-it head-his-the-young
   "Indeed, my life was more noisy than what his young head could tolerate".

<div dir="rtl">

إن حياتي هنا أكثر صخباً من أن يتحملها رأسه اليافع.

</div>

The second translation strategy utilizes negative verbs/verbals (11 instances, 20.75%) to relay the meaning of *too* negation. This is a familiar strategy in English as well as in Arabic to express negation implicitly rather than explicitly. For example, "the act of denying doing something" implies "the act of not admitting doing it". Consequently, this strategy constitutes an important option when translating negation in general and *too* negation in particular. The two examples below are illustrative:

9.　Anna dies or becomes too ill to continue writing it. (FS)

ʿaana maatat　　　　ʾaw balaɣat min-al-maraḍi　　　　ḥaddan ḥaala bayna-haa
Anna died　　　　or reached from-the-illness　　　　degree prevented between-her
wa-bana-l-ʾistimraari　　　fi-l-kitaabati
and-between-the-continuing　in-the-writing
"Anna died or reached a degree of illness that prevented her from continuing writing".

آنا ماتت أو بلغت من المرض حداً حال بينها وبين الاستمرار في الكتابة.

10.　It just seemed too good to be true that he was going to be rescued from the Dursleys. (HP/5)

fa-qad badaa ʾanna ʾinqaað̣a-hu min ʾaalii duurislii ʾamrun yafuuqu ʾaḥlaama-hu
so-? seemed that rescuing-him from family Duresleys matter exceeds dreams-his
"It seemed that rescuing him from the Duresleys was something that exceeds his dreams".

فقد بدا أن إنقاذه من آلي دوريسلي أمر يفوق أحلامه.

The implicit negation in (9) and (10) is achieved by the use of the negative verbs *ḥaala* [prevented] and *yafuuqu* [exceed], which both imply propositions employing explicit negation in Arabic.

The third translation strategy employs the degree of the relevant attribute (10 instances, 18.87%) as a marker of implicit negation by using the degree formulas *min + maṣdar* [verbal noun], for example, *from the-smallness* [i.e., too small] and *verb/adjective + ʿalaa* [on], for example, (*grow*) *old on* [i.e., too old]. Observe the two examples below:

11.　Fines for violations were too small to be effective, ... (ES)

fa-l-ɣaraamaat-u ʿan-il-muxaalafaati　　kaanat min-aṣ-ṣiɣari biḥayθu　　faqadat faaʿiliyyata-ha
so-the-fines from-the-violations　　　were from-the-smallness so　　lost(they) effect-their
"The fines for violations were so small that they lost their effect"

فالغرامات عن المخالفات من الصغر بحيث فقدت فاعليتها.

12.　"No, now I am too old to learn anything." (BF)

laa wa-ʾanaa alʾaana kabirtu ʿalaa taʿallumi ʾayyi šayʾin
no and-I now grew old on learning any thing
"No, I have grown (too) old to learn anything".

لا وأنا الآن كبرت على تعلم أي شيء.

Finally, we have the mixed bag, which includes mistranslations/under-translations that account for 16/53 instances (a full 30.19%) in the cases of implicit negation. The two examples below are illustrative:

13.　but it'll take too long to explain now. (HP/3)

laakinna-haa　　　qiṣṣatun　　　ṭawiilatun
but-it　　　　　story　　　　long
"but it was a long story".

لكنها قصة طويلة.

14.　It is still far too cold to undress at night. (BF)

laayazaalu-ṭ-ṭaqsu　　baaridan　　jiddan　　li-xalʿi-θ-θiyaabi　　fi-l-layli
still-the-weather　　cold　　very　　to-take off-the-clothes　in-the-night
"The weather is still very [too] cold to take off clothes at night"

لايزال الطقس بارداً جداً لخلع ثيابي في الليل.

*5.3. Translating -ly Negation*

Table 4 presents the translation strategies which Arabic translators employ when rendering English *-ly* negation.

**Table 4.** Frequency and percentage of translation strategies in rendering *-ly* negation.

| Strategy | Frequency | Percentage |
|---|---|---|
| Correct explicit negation | 12 | 18.75% |
| Under-translated explicit negation | 16 | 25% |
| Vernacular (ungrammatical) *bilkaad* | 20 | 31.25% |
| Correct implicit negation | 12 | 18.75% |
| Mistranslations/omission | 4 | 6.25% |
| Total | 64 | 100% |

The sample of negative adverbs (which is extracted from BF, FS, GC, and ES, to the exclusion of HP) includes 34 instances of *hardly*, 19 of *scarcely,* and 11 of *barely*, coming to a total of 64 instances. These adverbs share the fact that they communicate a negative orientation when used in English sentences. That is why they are often interchangeable, albeit they may be sensitive to normality conditions (i.e., one may sound natural in one context, while another may not). For example, *John was barely 17 when he joined college* is natural, whereas *John was scarcely 17 when he joined college* is not. In terms of translation, the focus is on relaying the negative orientation which is shared by all of them.

Table 4 above shows only two successful strategies the translators have employed in rendering *-ly* negation: (1) Correct explicit negation (12 instances, 18.75%), and (2) correct implicit negation (12 instances, 18.75%). The remaining cases go for: (1) Inappropriate vernacular *bilkaad* (20 instances, 31.25%), (2) under-translated explicit negation (16 instances, 25%), and (3) mistranslations (three instances) and omission (one instance), together 6.25%.

Let us start with cases where *-ly* negation is accounted for in Arabic (12/28 instances, 42.85%) using correct explicit negation. The following examples are illustrative:

15.　...this little kid who could barely walk ... (FS)
　　...haað-a-ṭ-ṭiflu-ṣ-ṣaʕiiru　　　'allaðii laa yakaadu yamšii ...
　　...this-the-kid-the-little　　　who not hardly walk ...
　"...this little kid who can hardly walk ... "

... هذا الطفل الصغير الذي لا يكاد يمشي ...

16.　The economics of society ... was hardly such as to provoke the curiosity of
　　a thoughtful man. (ES)
　　kaanat 'iqtiṣaadiyyaat-ul-mujtamaʕi ...　　takaadu laa tuθiiru fuḍuula　rajulin mufakkirin
　　were economics-the-society ...　　hardly not provoke curiosity　man thoughtful
　　"The economics of society ... could hardly provoke the curiosity of a thoughtful man".

كانت اقتصاديات المجتمع ... تكاد لا تثير فضول رجل مفكِّر.

The translators of (15) and (16) have successfully employed a negated *yakadu*, viz. *laa yakaadu* [not hardly] and *takaadu laa* [hardly not], which exactly capture the meaning of the *-ly* negation in them. One should note that *yakaadu* is an Arabic defective verb which translates into the negative adverb *hardly* when it is negated in Arabic, while it translates into *almost/nearly* when it is not negated, for example, كاد أن يسقط في البركة *kaada 'an yasquṭa fi-l-birkati* [almost (he) that fall in-the-pool] "He almost/nearly fell in the pool". The negated *yakadu* proves so useful when rendering English *-ly* negation.

However, *-ly* negation does not seem as straightforward as (15) and (16) may suggest. While capturing the notion of negation in general, almost 58% of the Arabic renderings (16/28 instances) fail to account for the nuance inherent in *-ly* negation. Instead, this kind of negation is erroneously

relayed as Arabic negation that corresponds to English negation by *not*, thus amounting to serious under-translations. Following are some illustrative examples:

17.    The children of large families hardly ever learn to talk to themselves aloud, … (BF)
 lam yataʿallam ʾawlaadu-l-ʿaaʾilaati-l-kabiirati    ʾan yataḥaddaθuu    ilaa ʾanfusihim
 not learned children-the-families-the-big    that talk    to themselves
 bi-ṣawtin masmuuʿin …
 with-voice audible …
 "The children of large families did not learn to talk to themselves in an audible voice … "

لم يتعلم أولاد العائلات الكبيرة أن يتحدثوا إلى أنفسهم بصوت مسموع.

18.    The Freifrau scarcely heeded her. (BF)
 lam tubaali-l-baaruunatu    bi-haa
 not heed-the-Baroness    with-her
 "The Baroness did not heed her".

لم تبالِ البارونة بها.

By way of illustration, in (17) the translator obliterates the subtle nuance of the negation in *hardly* by opting for explicit Arabic negation by *lam* [not]followed by the main lexical verb *yataʿallam* [learn], which back-translates into English negation by *not*, viz. *"The children of large families did not learn to talk to themselves … "*. To capture the negation inherent in *hardly*, one may need to employ a negated *yakaadu* viz. لا يكاد أولاد العائلات الكبيرة يتعلمون التحدث إلى أنفسهم بصوت مسموع *laa yakaadu* *ʾawlaadu-l-ʿaaʾilaati-l-kabiirati yataʿallamuuna-t-tataḥadduθa ʾilaa ʾanfusihim bi-ṣawtin masmuuʿin …* [not hardly children-the-families-the-large learn-the-talking to themselves with-voice audible] "The children of large families hardly learn to talk to themselves in an audible voice … ". Or, alternatively, one may use a paucity adverb like *naadiran maa* and *qalamaa* [rarely], which both inhere the nuance that "the circumstances in which those children live hardly allow them to talk to themselves", viz. نادراً ما/قلما يتعلم أولاد العائلات الكبيرة التحدث إلى أنفسهم بصوت مسموع *naadiran maa/qallamaa yataʿallamu* *ʾawlaadu-l-ʿaaʾilaati-l-kabiirati* at-*tataḥadduθa ʾilaa ʾanfusihim bi-ṣawtin masmuuʿin …* [rarely/hardly learn children-the-families-the-large the-talking to themselves with-voice audible] "Rarely/hardly (do) the children of the large families learn to talk themselves in an audible voice".

The second strategy for rendering -*ly* negation is the employment of the vernacular negative adverb *bilkaad*, which is a malformed version of the standard *la yakaadu*. The question is whether it is appropriate to use a vernacular form when it is possible to utilize the standard negated *yakaadu*. What is surprising here is the absence of this vernacular form in *Harry Potter's* translation where the informal register may sanction it and the frequency of using it in the other works—it accounts for 31.25% of the -*ly* data (20/64 instances). By way of illustration, the example in (31):

19.    I hardly know you, Augustus Waters. (FS)
 ʾanaa bilkaad    aʿrifu-ka    yaa ʾuʕusṭus wuutarz
 I hardly    know(I)-you    oh Augustus Waters
 "I hardly know you Augustus Waters".

أنا بالكاد أعرفك يا أوغسطس ووترز.

can readily be rephrased naturally in standard Arabic using the negated *yakaadu* as in (20) below.

20.    ʾanaa laa ʾakaadu    aʿrifu-ka    yaa ʾuʕusṭus wuutarz
 I not hardly(I)    know(I)-you    oh Augustus Waters
 "I hardly know you Augustus Waters".

أنا لا أعرفك أكاد يا أوغست ووترز.

Next, we have the strategy of implicit negation which accounts for 18.75% in the *-ly* negation data (12/64 instances). They mainly employ *paucity* or *difficulty* expressions in an attempt to capture the negative nuance inherent in *-ly* negation. Consider the two examples below:

21.  Auguste nowadays scarcely ever went out at all, ... (BF)
    fii miθli haaðihi-l-ʿayyaami      kaana      min-an-naadiri bi-nnisbati li-ʾuuʕast
    in like these-the-days      was      from-the-rarity as-regards to-Auguste
    ʿan taðhaba xaariji-l-manzili
    that go(she) outside-the-house
    "In these days Auguste rarely leaves home".

<div dir="rtl">في مثل هذه الأيام كان من النادر بالنسبة لأوغست أن يذهب خارج المنزل.</div>

22.  It is asked incessantly, most of the time however hardly noticeably, ... (BF)
    suʾaalun yusʾalu      bi-stimraarin      raʕma ʾanna-hu yulaaḥaðu      bi-ṣuʿuubatin ...
    question be asked      with-continuity      despite that-it be noticed      with-difficulty ...
    "A question asked incessantly despite (the fact that) it is noticed with difficulty ... "

<div dir="rtl">سؤال يسئل باستمرار رغم أنه يلاحظ بصعوبة ...</div>

In (21), the translator successfully employs a paucity expression *min-al-naadiri* [from the rarity] to capture *-ly* negation. One should note that a negated *yakaadu* can be readily used for that purpose, viz. <span dir="rtl">في مثل هذه الأيام لا تكاد أوغست تخرج من المنزل</span> *fii miθli haaðihi-l-ʿayyaami laa takaadu ʾuuʕast taxruju min-al-manzili* [in like these days Auguste not hardly goes out from-the-house] "In these days Auguste hardly ever goes out". In (22), a difficulty expression *bi-ṣuʿuubatin* [with difficulty] is utilized. The difficulty expression approximates rather than replicates *-ly* negation. A negated *yakaadu* would capture the meaning more closely, viz. <span dir="rtl">سؤال يسئل باستمرار رغم أنه لا يكاد يلاحظ</span> *suʾaalun yusʾalu bi-stimraarin raʕma ʾanna-hu laa yakaadu yulaaḥaðu* [question be asked with-continuity despite that-it not hardly be noticed] "A question (that is) asked incessantly, despite the fact that it is hardly noticed, ... ".

To close this section, let us examine two mistranslations (out of three) which are found in the *-ly* negation data (they all come from ES) below:

23.  This hardly seems like a particularly exciting subject for historical scrutiny.
    yakaadu      haaðaa      ʾašbaha      bi-mawḍuuʿin      muθiirin bi-wajhin xaṣṣin
    hardly      this      like      with-subject      exciting with-face particular
    lil-baḥθi-l-taariixiyyi
    for-research-historical
    "This almost seems like a particularly exciting subject for historical research".

<div dir="rtl">يكاد هذا أشبه بموضوع مثير بوجه خاص للبحث التاريخي.</div>

24.  This is particularly true when we begin at the stage of scarcely-better-than-subsistence ...
    wa-haaðaa      ṣaḥiiḥun      bi-wajhin xaṣṣin      ʿindamaa      nabdaʾu marḥalata
    and-this      true      with-face particular      when      begin(we) stage
    maa yakaadu yaziidu      ʿalaa      mujarradi-l-ʿayši ...
    which hardly more      than      mere-the-livelihood ...
    "This is particularly true when we begin the stage which is almost beyond mere livelihood ... "

<div dir="rtl">وهذا صحيح بوجه خاص عندما نبدأ مرحلة ما يكاد يزيد على مجرد العيش ...</div>

To explain, the translator in (23) wrongly uses the affirmed rather than the negated *yakaadu*, which is an approximating rather a negating marker (i.e., here it communicates the message that "X is almost Y"). To capture *-ly* negation, the translation should read <span dir="rtl">لا يكاد هذا يبدو صحيحاً لموضوع مثير بوجه خاص للبحث التاريخي</span> *laa yakaadu haaðaa yabduu šabiihan*



*li-mawḍuuʕin muθiirin bi-wajhin xaṣṣin lil-baḥθi-l-taariixiyyi* [not hardly this seems like with-subject exciting with-face particular with-the-research-historical] "This hardly seems like a particularly exciting subject for historical research ... ". In (24), the translator also fails to employ a negated *yakaadu*, perhaps misguided by the presence of *maa,* which coincides, in form, with a negative particle, but is used as a relativizing marker in this sentence. To correct this mistake, the negative particle *laa* needs to be inserted before *yakaadu* in order for the Arabic translation to read وهذا صحيح بوجه خاص عندما نبدأ مرحلة ما لا يكاد يزيد على مجرد العيش *wa-haaðaa ṣaḥiiḥun bi-wajhin xaṣṣin ʕindamaa nabda'u marḥalata maa **laa** yakaadu yaziidu ʕalaa mujarradi-l-ʕ ayši ...* , which corresponds to "This is particularly true when we begin the stage which hardly goes beyond mere livelihood ... ".

## *5.4. Translating Determiner Negation*

English determiner negation by *little* and *few* furnishes an utterance with a negative orientation just like *too* and *-ly* adverbial negation. They may also be used as adjectives to denote their lexical meaning by indicating smallness in size and number, respectively, which corresponds to *ṣaʕiir* [small/little] and *qaliil* [little/few] in Arabic. For example, there is not much beyond their semantics in *there are little children playing in the garden* and *The next few years will be prosperous*. However, *little* and *few* are often employed as negative determiners that contrast with their positive counterparts *a little* and *a few*. Compare "There is little time for discussion" لا يوجد إلا القليل من الوقت للنقاش *laa yuujadu 'illa-l-qaliilu min-al-waqti li-n-niqaaši* [not exist except-the-little from-the-time for-the-discussion] with "There is a little time for discussion" يوجد بعض الوقت للنقاش *yuujadu baʕḍu-l-waqti li-n-niqaaši* [exist some-the-time for-the-discussion] and لا يوجد إلا القليل من الأخطاء في التقرير *laa yuujadu 'illa-l-qaliilu min-al-'axṭaa'i fi-t-taqriiri* [not exist except-the-few from-the-mistakes in-the-report] with *There are a few mistakes in the report* يوجد بعض الأخطاء في التقرير *yuujadu baʕḍu-l-'axṭaa'i fi-t-taqriiri* [exist some-the-mistakes in-the-report]. While *little* and *few* color the utterances with a negative orientation, *a little* and *a few* color it with a positive orientation, hence the different Arabic renderings. The discussion in this section aims to show to what extent Arabic translators are aware of this subtle type of negation.

### 5.4.1. Determiner Negation by *Little*

Table 5 below displays the translation strategies that the translators have followed in dealing with *little* determiner negation.

**Table 5.** Frequency and percentage of translation strategies for *little* determiner negation.

| Translation Strategy | Frequency | Percentage |
|---|---|---|
| Correct explicit negation | 33 | 38.38% |
| Incorrect implicit negation | 28 | 32.50% |
| Under-translated explicit negation | 18 | 20.95% |
| Correct implicit negation | 7 | 8.14% |
| Total | 86 | 100% |

Out of the 99 extracted examples featuring *little*, 86 (86.87%) are found to involve a negative orientation that goes beyond its denotative (dictionary) meaning. The examination of *little* negation data shows that the translators' attempt to handle this kind of negation involves four strategies: proper explicit negation (33 instances, 38.37%), under-translated explicit negation (18 instances, 20.93%), proper implicit negation (seven instances, 8.14%), erroneous implicit negation (25 instances, 29%), and mistranslation (one instance (1.16%).

To start with the first category, where determiner negation is rendered by explicit negation, which is the most frequent (33 instances, 38.37%), one can notice two main translation strategies. The first (23 instances/69.70%) usually employs explicit negation, with the exception particle *'illaa* or *siwaa* [except]

followed by a paucity-derived word, for example, *'illaa qaliilan* [except little/few] or *siwaa-l-qaliila* [except-the-little/few]. Consider the following example:

25.     We all know that there's very little time. (TH)
        kullu-naa na'rifu    'anna-hu           laysa ladaynaa siwaa qaliilun min-al-waqti
        all-we know        that-it             not have-we except little from-the-time
        "We all know that we don't have except little time" (i.e., "we all know that we only have little time").

<div dir="rtl">

كلنا نعرف أنه ليس لدينا سوى قليل من الوقت.

</div>

The second translation strategy, which claims (10 instances, 30.30%), employs a negated antonym, for example, *lan … al-kaθiira* [not … a lot/much], as is shown in the following example:

26.     And all with very little effort on your part, I assure you. (HP/5)
        wa-lan yakuuna     'alay-ka 'an tabðula-l-kaθiira min-al-waqti     'utam'inu-ka
        and-not be          on-you that spend-the-much from-the-time     assure (I)-you
        "And you won't have to spend much time, I assure you".

<div dir="rtl">

ولن يكون عليك أن تبذل الكثير من الوقت أطمئنك.

</div>

One should note that Arabic can also employ *belittling* expressions such as *laa yuðkar* [not to be mentioned] or *laa yastaḥiq-uð-ðikr* [not worth mentioning] to capture the meaning of *little*-negation in examples like (26), which can be rephrased in (27):

27.     wa-lan yakuuna     'alay-ka baðlu juhdin yastaḥiqqu-ð-ðikra     'utam'inu-ka
        and-not be          on-you making effort worth-the-mentioning     assure(I)-you
        "And you won't have to make (any) effort worth of mentioning, I assure you".

<div dir="rtl">

ولن يكون عليك بذل جهد يستحق الذكر أطمئنك.

</div>

The second translation strategy (18 instances, 20.93%) includes cases where the translator succeeds in recovering Arabic explicit negation but, unfortunately, misses the focus of determiner negation (i.e., he/she under-translates this subtle type of negation). By way of illustration, witness the following example:

28.     The truth was, I had very little idea how dangerous things were. (TH)
        fi-l-ḥaqiiqati        lam       'akun 'a'rifu madaa xuṭuurati-l-'amri
        in-the-fact        not       be know(I) extent dangerousness-the-situation
        "In fact, I didn't know how dangerous the situation was".

<div dir="rtl">

في الحقيقة لم أكن أعرف مدى خطورة الأمر.

</div>

As can be seen, *little*-negation in (28) is rendered in Arabic to what corresponds to *not* negation in English, thus missing the nuance of this type of negation. To capture this nuance, the Arabic rendering should employ an *exception* expression along with explicit negation as in (29):

29.     fi-l-ḥaqiiqati     lam 'akun 'a'rifu     'illa     'aqali-l-qaliili      'an
        in-the-fact      not be know(I)       except     smallest-the-small     about
        madaa xuṭuurati-l-'amri
        extent dangerousness-the-situation
        "In fact, I knew only little about how dangerous the situation was".

<div dir="rtl">

في الحقيقة لم أكن أعرف إلا أقل القليل عن مدى خطورة الأمر.

</div>

The next two strategies involve the translator's attempt to render *little* negation by implicit negation. The outcome is far from being impressive: Only seven cases (8.14%) may be considered

successful in implementing this strategy, while 28 cases (32.50%) falter in this respect. Following are two examples where the first succeeds in relaying *little* negation (30), while the second falters (31):

30.　　"You know that Father punishes you very little", said Sidonie coaxingly. (BF)
　　　　'anta ta'rifu yaa birnaard 'anna 'abii naadiran maa yu'aaqibuka ...
　　　　you know oh Bernard that father(my) rarely punish(you)
　　　　"You know Bernard that my father rarely punishes you ... "

<div dir="rtl">أنت تعرف يا بيرناد أن أبي نادراً ما يعاقبك ...</div>

31.　　Very little has changed with her health. (TH)
　　　　laqad　　　ṭara'a　　　　taḥassunun 'alaa ṣiḥḥati-haa
　　　　?　　　　happened　　　improvement on health(her)
　　　　"Her health has improved".

<div dir="rtl">لقد طرأ تحسن على صحتها.</div>

In (30), the translator employs the paucity expression *naadiran maa* [rarely] to capture the meaning of *little* negation. By contrast, the Arabic rendering in (31) embraces a positive orientation towards the referent's health conditions, which runs counter to the negative orientation in the English utterance. To capture this orientation, the Arabic rendering may be rephrased using explicit negation along with a *belittling* expression as in (32) below:

32.　　lam yaṭra'　　　taḥassunun　　　yastaḥiqqu-ð-ðikra　　　'alaa　　　ṣiḥḥati-haa
　　　　not happened　　improvement　　worth-the-mentioning　　on　　　health-her
　　　　"There wasn't any improvement worth mentioning about her health".

<div dir="rtl">لم يطرأ تحسن يستحق الذكر على صحتها.</div>

### 5.4.2. Determiner Negation by *Few*

Table 6 below presents the frequency and percentage of translation strategies which the translators have employed in rendering *few* determiner negation.

**Table 6.** Frequency and percentage of translation strategies for *few* determiner negation.

| Translation Strategy | Frequency | Percentage |
|---|---|---|
| Incorrect implicit negation | 12 | 60% |
| Correct explicit negation | 7 | 35% |
| Correct implicit negation | 1 | 5% |
| Total | 20 | 100% |

Negation by *few* is the least frequent in the corpus. Out of 93 extracted examples involving the employment of *few*, only 20 are found to furnish the English utterance with a negative orientation. In the rest of the examples, the determiner *few* reflects its dictionary meaning, which corresponds to *qaliil* [few] in Arabic, without any coloration of negation. In such cases, the rendering of *few* into Arabic is straightforward, as no negation is to be accounted for. The following example is illustrative:

33.　　The next few weeks is real important for Mae Mobley. (TH)
　　　　kaanat-il-'asaabii'u-l-qaliilatu-t-taaliyatu　　haamatan jiddan binnisbati　　'ilaa　　maw muublii
　　　　were-the-weeks-the-few-the-next　　　　　　important very as regards　　　to　　Mae Mobley
　　　　"The next few weeks were very important for Mae Mobley".

<div dir="rtl">كانت الأسابيع القليلة التالية هامة جدا بالنسبة إلى ماو موبلي.</div>

However, when *few* is employed as a negative determiner, which is meant to express the producer's unfavorable attitude towards the state of affairs in question, explicit Arabic negation may be needed. The data shows that explicit negation has been correctly employed in seven instances (35%), while implicit negation is erroneously employed in 12 instances (60%), and only once correctly (5%). The two examples below are representative of the success and failure in rendering *few* negation:

34. They stopped at no inns, and exchanged very few words. (BF)

| lam yatawaqqafaa | ʿinda | ʾayyi ḥaanatin | wa-lam yatabaadalaa | ʾilla-l-qaliilu |
|---|---|---|---|---|
| not stopped(dual) | at | any inn | and-not exchanged | except-the-few |

min-al-kalimaati
from-the-words
"They didn't stop at any inns and didn't exchange but few words".

<div dir="rtl">لم يتوقفا عند أي حانة ولم يتبادلا إلا القليل من الكلمات.</div>

35. And there are as yet few business models for journalism originating on the Internet. (GC)

| wa-hunaaka | ḥatta-l'aan | ʿadadun qaliilun | min namaaðiji-l-'aʿmaali-ṣ-ṣaḥafiyyati |
|---|---|---|---|
| and-there | till-now | number small | from models-the-businesses-the-journalism |

an-naaši'ati　　　ʿala-l-intarnit
the-new　　　　on-the-Internet
"There are as yet a small number of new business models on the Internet".

<div dir="rtl">وهناك حتى الآن عدد قليل من نماذج الأعمال الصحفية الناشئة على الإنترنت.</div>

As is clear in (34) above, the translator has duly accessed explicit negation along with an *exception* expression in Arabic to account for *few* negation. By contrast, the translator has failed to invest explicit negation and, consequently, opts erroneously for an affirmative utterance in (35). The competent reader can readily feel the missing negative orientation in the rendering due to the translator's failure to capture the pragmatics of *few* negation. To remedy this situation, explicit Arabic negation along with an *exception* expression may be accessed to furnish a negative orientation (36 below):

36.

| wa-laa | yuujadu | ḥatta-l'aan | ʾillaa ʿadadun qaliilun min namaaðiji-l-'aʿmaali-ṣiḥḥati-haa |
|---|---|---|---|
| an-laa | exist | till-now | except number small from models-the-businesses |

ṣ-ṣaḥafiyyati　an-naaši'ati　　　　　ʿala-l-intarnit
the-journalism　the new　　　　　　the-journalism
"There aren't as yet but few new business models on the Internet".

<div dir="rtl">ولا يوجد حتى الآن إلا عدد قليل من نماذج الأعمال الصحفية الناشئة على الإنترنت.</div>

## 6. Discussion

In response to the first research question concerning how Arabic translators tackle *too* adverbial negation (which is formally missing in Arabic) and what translation strategies they employ, the study shows that they resort to explicit negation and implicit negation as two general strategies. This clearly proves that the recovery of Arabic explicit negation is an effective translation strategy in dealing with *too* negation. Within explicit negation, three strategies are employed: unpacking by coordination, nominalization, and indicating degree of attribute.

Unpacking by coordination turns out to be a very effective translation strategy for rendering *too* negation. The translators have successfully managed to explicate the English negation encapsulated in the negative marker *too* by using a consequential coordinate clause involving explicit negation. In this way, for example, the *too* negation in *Harry was too deeply asleep to hear her* is successfully relayed into an Arabic rendering that back-translates into *Harry was deeply asleep, so he didn't hear her*, which is an alternative English textualization that employs generic explicit negation. Arabic translators; therefore, need to be fully aware of this workable strategy when dealing with *too* negation.

Resorting to explicit negation by nominalizing the English verb in a simple or complex Arabic structure also proves to be a workable translation strategy for rendering *too* negation. For example, the translator has managed to recover Arabic generic negation by nominalizing the English verb *save* into the Arabic verbal noun *'inqaaði* "saving", in *I was too late to save the girl,* into a simple Arabic structure which literally back-translates into *\*I couldn't saving the girl.* Similarly, the translator has successfully nominalized the verb *drive away,* in *but his Patronus was too feeble to drive the dementor away,* in the English infinitive clause, into *'ibʕaadi* (driving away) in an Arabic complex structure (which back-translates into a workable English textualization, viz. *but his Patronus wasn't strong enough to drive the dementor away*).

Indicating the degree of the attribute in question, for its part, presents itself as a very useful translation strategy when rendering *too* negation by explicit Arabic negation. It corresponds to an awkward English textualization that may paraphrase *too* negation. For example, the Arabic idiomatic rendering of *Professor Trelawney seemed too tipsy to have recognized Harry* back-translates into the awkward English paraphrase *Professor Trelawney was drunk to the extent that she didn't recognize Harry.*

There are few cases when the translator's recovery of Arabic generic negation does not convey the nuance of *too* negation properly. For example, the *too* negation in *but Harry was too used to this to care* is rendered into Arabic explicit negation that back-translates into *but Harry didn't care about this*, thus doing away with the shade of meaning inherent in *too* negation. To capture this shade of meaning, the translator could have indicated the degree of the attribute in question by offering an Arabic rendering that back-translates into the English paraphrase *but Harry was used to this to the extent that he didn't care* or, alternatively, the translator could have unpacked *too* negation by coordination by offering what back-translates into *Harry was used to this, so he didn't care.*

As for implicit negation, it is not as successfully employed as explicit negation. However, this does not mean that implicit negation is not a viable option; it just suggests that it needs to be utilized more carefully by calling up three strategies: use of comparative form, use of negative, and indication of degree of attribute, which prove to be effective in rendering *too* negation.

The employment of the comparative form idiomatically renders many cases of *too* negation. For instance, the English *too* negation in *because it was too long to memorize* is correctly relayed into Arabic to what back-translates into an English workable textualization, viz. *because it was longer than it could be memorized.* Similarly, the use of negative verbs/verbals may appropriately capture *too* negation. The English *too* negation in *Anna dies or becomes too ill to continue writing it*, for example, lends itself to translating by employing a negative verb whose semantics takes care of negation, viz. the Arabic rendering back-translates into *Anna died or reached a degree of illness that prevented her from continuing writing.* Likewise, employing set formulas to indicate the degree of the relevant attribute succeeds in capturing *too* negation implicitly. Notice how the Arabic rendering, whose back-translation is *the fines for violations were so small that they lost their effect,* proves to be an idiomatic translation of the *too* negation in *Fines for violations were too small to be effective.*

Despite the successful employment of these strategies, the translators have failed to use implicit negation correctly in about one third of the cases. This makes the strategy of implicit negation more challenging than that of explicit negation. Therefore, the translator's first strategy should be to consider explicit negation, and he/she needs to exercise utmost care when opting for implicit negation. For example, the *too* negation should not be confused with the negation-free intensifier *jiddan* "very". In this way, the *too* negation in *It is still far too cold to undress at night* should not be rendered in Arabic as لايزال الطقس بارداً جداً لخلع الثياب في الليل *laayazaalu-ṭ-ṭaqsu baaridan jiddan li-xalʕi-θ-θiyaabi fi-l-layli* "*\*The weather is still very cold to take off clothes at night.*" To use Arabic implicit negation properly, the translator could have employed an Arabic negative verb whose semantics takes care of *too* negation, viz. كان الجو البارد يمنعنا من خلع ملابسنا في الليل *kaan-al-jawwu-l-baaridu yamnaʕu-naa min xalʕi malaabisi-naa fi-l-layli* "The cold weather prevented us from taking off our clothes at night."

To address the second research question concerning the translation strategies in rendering *-ly* negation, the data shows that the successful use of explicit negation and implicit negation accounts

for only (37.50%) in the corpus. The rest of the cases goes for the vernacular *bilkaad* (31.70%), which is ungrammatical, and mistranslations/under-translations/omission (31.75%). This clearly shows how problematic rendering -*ly* negation into Arabic is. To render this type of negation properly, translators into Arabic need to be made aware of the negated Arabic verb *yakaadu* when employing explicit negation, as well as *paucity/difficulty* expressions when opting for implicit negation. Only then can the pragmatics of -*ly* negation be captured in Arabic.

Arabic negation by explicit negative particles emerges as the most common translation strategy for rendering the -*ly* negative adverbs, which clearly indicates the translators' awareness of their negative orientation. However, the coding of this orientation in Arabic seems to be a challenging task. In fact, more than half involves under-translating this adverbial negation by rendering it into what corresponds to negation by *not* in English. In this way, the subtle nuance of this type of negation is lost in translation.

To employ explicit negation properly when dealing with -*ly* negation, the translator needs to access the Arabic negated verb *yakaadu*, which does not seem to be an easy task. Apparently, Arabic translators more often than not fall in the trap of under-translation by offering English generic negation by *not*, viz. *The Barn didn't heed her* (back-translation of Arabic translation) for *Frefrau scarcely heeded her*. In this way, they fail to call up the appropriate negated *yakaadu*, viz. لم تكد البارونة تبالي بها *lam takad al-baaruunatu tubaali bi-haa* "The Baroness scarcely/hardly heeded her". Alternatively, they are erroneously attracted by the vernacular *bilkaad*, which is a negative adverb commonly used in most Arabic vernaculars for this type of negation. This option reflects the translator's deficient knowledge of Standard Arabic, which has its standard version (the negated *yakaadu* verb) for the vernacular *bilkaad*. Arabic translators; therefore, need to be cautioned against rendering *ly*-negation into the generic *not* negation as well as the use of vernacular *bilkaad*.

Arabic implicit negation may also be considered when translating -*ly* negation. When implementing this translation strategy, the translator needs to call up Arabic *paucity* expressions such as *naadiran maa/qallamaa* "rarely" or the difficulty expression *bi-ṣuʕuubatin* which approximates than replicates -*ly* negation implicitly. Though implicit negation is used much less frequently than explicit negation where several errors are made, it proves to be very appropriate in some cases, especially when the translator utilizes *paucity* expressions.

To respond to the third research question regarding the rendering of *little* determiner negation, results show that the success rate is less than 50%, which clearly indicates how problematic *little* negation is when relaying it into Arabic.

The most effective translation strategy when employing explicit negation is to use *exception* expressions combined with *paucity* words, which can properly capture this kind of subtle negation. Arabic translators; however, need to be cautioned against falling in the trap of under-translating *little* negation into what corresponds to English *not* negation, viz. *The truth was, I had very little idea how dangerous things were* may wrongly be translated into an Arabic rendering that back-translates into *In fact I didn't know how dangerous the situation was*, a mishap which belies several cases of *little* negation. To capture *little* negation here, the translator needs to use explicit negation along with an *exception* expression, viz. في الحقيقة لم أكن أعرف إلا أقل القليل عن مدى خطورة الأمر. *fi-l-ḥaqiiqati lam ʾakun ʾaʕrifu ʾilla ʾaqalli-l-qaliili ʕan madaa xuṭuurati-l-ʾamri*, which back-translates into *In fact I knew only very little about how dangerous the situation was*.

Negated *antonyms* may also be used to relay *little* negation explicitly, though at a lesser degree of success, viz. *And all with very little effort on your part, I assure you* relatively corresponds to the Arabic rendering that back-translates into *And you won't have to spend much effort, I assure you*. However, the employment of explicit negation with a *belittling* expression such *laa yastaḥiqu-ið-ðikr* "not worth mentioning" can be a more effective strategy for capturing the shade of meaning in *little* negation than using explicit negation with a negated antonym, viz. ولن يكون عليك أن تبذل الكثير من الوقت أطمئنك *wa-lan yakuuna ʕalay-ka ʾan tabðula-l-kaθiira min-al-waqti ʾuṭam'inu-ka*, which back-translates into *And*

*you won't have to make a lot of effort, I assure you*. Consequently, the Arabic translator needs to be alerted to the importance of using *exception* expressions as well as *belittling* expressions when using explicit negation in rendering *little* negation.

As for implicit negation, which is employed in only few cases, it resorts to expressions inherently marked for paucity such as *naadiran maa/qallamaa* "rarely" independently of negative particles. The bulk of cases; however, indicates that the translators are only little aware of the negative orientation furnished by *little* negation. In fact, the erroneous translations furnish a positive rather negative orientation of *little* negation. For example, the rendering of *Very little has changed with her health* in لقد طرأ تحسن على صحتها *laqad ṭara'a taḥassunun ʿalaa ṣiḥḥati-haa* into *Her health has improved* has completely deprived the Arabic translation of the negative orientation. It is of utmost importance; therefore, to alert Arabic translators to the need to recover explicit negation along with *paucity* or *belittling* expressions to capture *little* negation. In this way, the above example should be rendered as لم يطرأ تحسن يذكر على صحتها *lam yaṭra' taḥassunun yuðkaru ʿalaa ṣiḥḥati-haa*, which back-translates into *There wasn't any improvement worth mentioning with her health.*

Finally, in response to the fourth research question, the results show that *few* determiner negation proves to be so problematic in Arabic translation. This finding clearly points to the challenging subtlety of *few* negation which needs to be brought to the consciousness of Arabic translators who are supposed to be aware of the difference between *few* as a negative operator and *few* as a mere determiner. Apparently, the translators are not aware of the need to recover explicit Arabic negation along with an *exception* expression to relay the negative attitude encapsulated in this type of subtle negation. The attempt to employ implicit negation has failed except in one instance, which clearly indicates that explicit rather than implicit negation is the appropriate translation strategy to use when rendering *few* negation. Translators into Arabic; therefore, need to be sensitized to the nature of *few* negation and the appropriate strategies that may be used to render its pragmatics.

## 7. Conclusions

Tables 2–6 report my judgments about the success and competence of translators whose work I examined for this study. The findings show evidence that the pragmatics of this type of English negation is a challenging task in translation activity. Arabic translators, both professionals and more so student translators, need to be alerted to the fact that this type of negation, which formally does not exist in Arabic (and probably in several other languages), calls, in the first place, for recovering generic negation and, in the second place, for appropriately investing implicit negation in Arabic. Only then can the negative orientation, with which adverbial and determiner negation furnishes English utterances, be captured in Arabic translation.

In response to the research questions, the study has uncovered a rich spectrum of translation strategies that Arabic translators may employ to deal with non-generic English adverbial and determiner negation, which is found to be a problematic area representing a textual mismatch between English and Arabic. The translation strategy of recovering generic explicit negation in Arabic presents itself as a first priority that cuts across all types of non-generic adverbial and determiner negation. For its part, implicit negation may also cut across all types of non-generic adverbial and determiner negation, but to a lesser extent, when rendering it into Arabic. Since it is semantically based, it seems to be more challenging to Arabic translators.

**Funding:** This research was supported by Kuwait University, Research Grant No. [AE03/16].

**Conflicts of Interest:** The author declares no conflict of interest.

## Appendix A. List of Arabic Phonetic Symbols

/b/ voiced bilabial stop

/m/ bilabial nasal

/f/ voiceless labio-dental fricative

/ð/ voiced interdental fricative

/ð̠/ voiced interdental emphatic fricative

/θ/ voiceless interdental fricative

/d/ voiced alveolar stop

/t/ voiceless alveolar stop

/ḍ/ voiced alveolar emphatic stop

/ṭ/ voiceless alveolar emphatic stop

/z/ voiced alveolar fricative

/s/ voiceless alveolar fricative

/ṣ/ voiceless alveolar emphatic fricative

/n/ alveolar nasal stop

/r/ alveolar rhotic liquid

/l/ alveolar lateral liquid

/š/ voiceless alveo-palatal fricative

/j/ voiced palatal affricate

/y/ palatal glide

/w/ labio-velar glide

/g/ voiced velar stop

/k/ voiceless velar stop

/γ/ voiced uvular/post-velar fricative

/x/ voiceless uvular/post-velar fricative

/q/ voiceless uvular stop

/ʻ/ voiced pharyngeal fricative

/ḥ/ voiceless pharyngeal fricative

/ʼ/ glottal stop

/h/ voiceless laryngeal fricative

/i/ high front short vowel

/u/ high back short vowel

/a/ low half-open front-to-centralized short vowel

/ii/ high front long vowel

/uu/ high back long vowel

/aa/ low open front-to-centralized long vowel

/ee/ mid front long vowel

/oo/ mid back long vowel

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
