# Peer review of "English Adverbial and Determiner Negation: A Problematic Area for Arabic Translators"

_languages, doi:10.3390/languages4010017_

Reviewer 1 Report

A quick check for style/language; nothing serious.

Very minor issues: 

1)      Title and the paper: Arabic translators can be used if the reference is to translating into Arabic, which is the case any way. Perhaps, Arab translators may function better in terms of translators deciding on how to handle a particular language issue from English into Arabic.

2)      Abstract: Perhaps add the number of examples in the data discussed (270) and an indication of the genre/discourse (literary).

Author Response

Reviewer 1: (Two minor comments)

1.       I’d rather keep ‘Arabic translators’ instead of the suggested ‘Arab Translators’ in the title of the paper. The paper relates to all translators who work from English into Arabic, regardless of whether they are Arab or non-Arab.

2.       The suggested info is added to the Abstract.

Reviewer 2 Report

English Adverbial and Determiner Negation: A Problematic Area for Arabic Translators

Languages

1.      This is a good article that addresses a topic not usually discussed in Arabic translation theory and practice. It uses adequate corpus of works translated into Arabic to look at problems and strategies of translating negation. It does bring some knowledge to translation trainers and practitioners. There are, however, some issues that need to be taken care of to be a publishable paper.

2.      Arabic explicit negation particles are not well introduced and the authors misses one important one, maa. The time and aspect of some of these particles are significant for translation. For maa, when used with simple past, it is an absolute negation. Whereas lam, it usually corresponds to present perfect in negating a present-related past.

3.      On page 3, line 82, “implicit Arabic negation uses grammatical devices such as interrogatives and conditionals (A list of Arabic phonetic symbols in provided in Appendix 1)”. One would expect a list of those grammatical devices be given here with some examples. The note about the Arabic phonetic symbols is irrelevant here.

4.      Table 1 needs to be revisited. Figures presented there are confusing. The table caption is “Distribution of English negative markers in the corpus” without making a differentiation between explicit and implicit markers. While the figures and thus percentages are descending from the highest, we find them go up again in row 5. This is because they were divided in two groups, but pooled in one table. Frequency is a statistical issue and needs to be presented accordingly. Otherwise, the table should be split in two tables, each for a different type of markers. The same rearrangement needs to be done for the rest of tables.

5.      Table 2 seems to suffer from mispresentation of figures. Row 4, point C states that this strategy accounts for 11 cases making 23.4%. However, in the narrative description of the statistics, the discussion refers to 12 cases (25.53%). There is, of course, inconsistency in presenting the findings. Also, the last row, which has its own frequency and percentage is not commented on in the narrative.

6.      Table 2 needs to be reformatted by removing the first row. As the table presents translations strategies, including the frequency of explicit negations in the source texts is not right. It is better to move that to the discussion and keep the table for translation strategies only. This also applies to Table 3.

7.      Repeating the figures right after the table is not helpful. It is better to state the figures at the beginning of the relevant paragraphs that discuss each translation strategy.

8.      For reference purposes, the author gives between brackets at the end of each example a reference code to the work from which it is taken such HP 1, BF and FS. This need to be supplemented by the page number of the example in both the source text and the target text. However, these codes do not mean much to the reader unless they are added next to each work in the section of Textual Material, for example Harry Potter and the Sorcerer's Stone (HP 1).

9.      In the translation of example 2 (p 8, line 225) suggested by the author, there seems to be shift of focus:

Harry was too deeply asleep to hear her. (HP/4)

lam yasmaʻ-haa haarii li'anna-hu kaana ɤaariqan fii nawmi-hi

"Harry didn't hear her because he was deeply asleep'.

10.  One page 5, line 130 “Al-Ghazalli (2013) discusses the translation of Arabic implicit negation in a poorly-written paper. Apart from the poor quality of this study.” The critique here seems not well situated and formulated. The poor quality could be highlighted indirectly and the main relevant points be presented. It is worth mentioning that in the reference list, it is spelled al-Ghazali (with one l).

11.  Page 3, lines 85-87: “Dendane and Dendane (2012), however, indicate that this diversity in generic negative particles can be problematic for machine translation.” seems irrelevant statement in this article since the discussion is not about machine translation.

12.  The section of discussion mostly repeats what has been given in the section of analysis. The section on discussion needs to remove repeated figures and statements, or better, it should be merged with the analysis section with its own new discussion and statement given together at the end of the section in a way of wrapping up the analysis.

13.  Table 4, the use of “Inappropriate vernacular bilkaad”. This is a malformed version of la yakaad. It is a similar case of takaadu laa (p 14), swapping the places of the expression components makes it ungrammatical. Many translators and speaker of Arabic are unware of this issue. For the translator, both are standard language. This need to be highlighted and explained. For the second, it has to be singled out as ungrammatical as well.

14.  Table 5 needs reformatting where frequencies should be descending. It does not matter what kind of strategy is used and whether it is correct or incorrect. Row 5 with 28 cases (32.5%) should come second. The same applies to Table 6. Also, the mathematic in Table 6 is incorrect. The findings and discussion should be updated accordingly.

15.  The transliterations of Arabic examples do not follow a particular scheme. It is hard to follow and sometime misleading. Why not providing the examples in Arabic script? The transliterated form could also be given next to it according to a clear system such as the ISO’s.

16.  Proofreading should be performed:

The repetition of ‘that’ on page 2, line 33: “However, there are only very few studies that that have dealt with Arabic and English negation from a translational perspective”, and the spelling of “translatiing’ on p 24, line 674.

17.  The Conclusion section should be more general without specification of tables, figures, number of examples etc.

References and in-text citation:

18.  Some sources in Arabic by Arabic authors and published in the Arab world such as Al-Makhzumi (2005) Anees (1975) and Nahr, (2004). are included in the non-Arabic section of the sources. They should be included in the Arabic section with their Arabic titles. For example,

أنيس، ابراهيم (1975). من أسرار اللغة. القاهرة: مكتبة الأنجلو المصرية.

If these sources are to be included in the reference list of scholarly list of referneces anyway, they need to be transliterated with an English translation between brackets, e.g.

Anees, Ibriham (1975). Min asraar allugha (from the secrets of language). Cairo: Anglo Egyptian Library.

19.  Names of Arab authors with the definite article al are inconsistently spelled: al-Ghazali, Al-Makhzumi, and AlSalem, unless these are the way they are spelled in internationally published works.

20.  If a citation is made from an Arabic source, ‘my translation’ should be added. Alternatively, a note is made stating that citations from Arabic are translated by the author.

21.  The referencing style is not consistent with first names of authors are given both as an initial as well as fully. The style adopted by the journal should be followed.

22.  Some sources stated in the article are not listed in the references: Fizgerald, 1996 (translated by Ali Suleiman, 2015); Heilbroner 1962 (translated by Rashid Al-Barrawi, 1976).

23.  In the Arabic section, Arabic translations are wrongly provided under the name of the translators. These should be listed under their authors’ names with the translators are credited. The publication year of the Arabic version is the relevant one. For example:

غرين، جون (2015). ما تخبئه لنا النجوم. ترجمة أنطوان باسيل. بيروت: شركة المطبوعات للتوزيع والنشر.

24.  While italics are used in some referencing styles to indicate titles, to differentiate them, Arabic titles are hardly the case; they do not look different. Italics in Arabic are not the standard style for titles anyway. The standard style for titles are to be underlined.

25.  Phonetic symbols of Arabic require representative examples (words) in English where possible to help non-Arabic speakers identify them. In case another transliteration scheme is followed instead, the same will be needed.

Author Response

Reviewer 2:

1.       (Notes 2 and 3) have been taken care of on pp. 2,3 and 5.  One should note that lam and maa  have nothing to do with Aspect in Arabic, as the reviewer mentions. They both negate past actions. The only difference has to do with frequency (lam being more frequent is Modern Standard Arabic).

2.       (Note 4) Table 1 on p. 8 only lists non-generic English explicit negative markers (Both adverbial and determiner markers) along with frequency and percentage. It has nothing to do with implicit negation as the reviewer mentions. The paper does not deal implicit negation in the English corpus in any way. All the data involve non-generic explicit negation (adverbial and determiner). Therefore, nothing needs to be done there. The Table just presents the facts in the English corpus.

3.       (Note 5) percentage corrected on p. 10 and one paragraph (pp. 10 and 11) added to cover the last category in Table 2 as suggested by reviewer.

4.       (Note 6)  Done (First row in Tables 2 and 3 removed)

5.       (Note 7) Done (Figures removed from paragraph following tables).

6.       (Note 8) I don’t think page numbers are needed.   

7.       (Note 9) Reviewer’s point (shift of focus) added.

8.       (Note 10) Text rephrased as the reviewer wants (p. 5)

9.       (Note 11) p. 4 (lines 96 and 97) Sentence deleted as suggested by reviewer.

10.   (Note 12) Figures and some statements are removed from Discussion section (pp. 24-29) as suggested by reviewer. 

11.   (Note 13) bilkaad amended as suggested by reviewer (pp. 17 and 27)

12.   (Note 14)  Tables 5 and 6 (amended to descending order of percentages) as suggested (pp. 20 and 23). Also, total corrected in Table 6 (p. 23).

13.   (Note 15)

14.   (Note 16) Done!

15.   (Note 17) Conclusion is made more general as suggested by reviewer.

16.   (Note 18) Arabic sources moved to Arabic references as requested.

17.   (Note 19) Consistency of Arabic article maintained.

18.   (Note 20) Done.

19.   (Note 21) Initials of first names used.

20.   (Note 22)  references added.

21.   (Note 23) Done.

22.   (Note 24) Done.

23.   (Note 25) Phonetic symbols are mainly IPA (List of phonetic symbols) / Also Arabic script is added to Arabic examples throughout the paper as suggested by reviewer.

Reviewer 3 Report

The author's statement "The findings reported in Tables 2-6 support the predictive title of this study that non-generic adverbial and determiner negation is a problematic area for Arabic translators" (ll 779-780) should be rephrased as, 'Tables 2-6 report my judgments about the success and competence of translators whose work I examined for this study.'  The extensive use of "properly" underscores the judgmental nature of the authorial voice, which substitutes evaluation for discussion.  We mostly read about translators' success and failure, not a discussion about the range of syntactic constructions (there is very little of this and it is not developed enough to be clear to a general audience) and lexical choices.  In addition, the negative structures examined here have a quantitative aspect to them, and in my opinion this aspect needs to be brought into the discussion. This would make the article interesting and original. As it is, it reads like a teacher's grading of his/her students' work. 

Many typos/errors in Arabic transliteration, to name a few, lines 77-78, 101, 103, 127, 367, 372, 722 ..

Need to specify that you limit your discussion to modern formal Arabic.  And please note that bi-l-kaad is *not* limited to what you call 'vernacular'--it is a full-fledged member of formal Arabic. 

 Typos and proper noun misspellings in English.  And by the way al-baaruuna is feminine in Arabic, which means that it should be rendered Baroness in English.

Tables 5 and 6 and the discussion of them are not clear, and there are some problems with the numbers reported.

A bit of unnecessary repetition in the first few pages, repeated quote and references ll 27-28, 76; 29-30, 79-81.

Author Response

Reviewer 3:

1.       Typos in transliteration corrected.

2.       Modern Standard Arabic is specified as the target language in this study. Note that the reviewer claims that bilkaad (which is designated as vernacular) is formal Arabic, which is not true. This also contradicts what the second reviewer requested, i.e. to dub it as ‘ungrammatical’ (which I did), not only vernacular.

3.       All the problems in tables have been corrected as requested by Reviewer 2 (including tables 5 and 6).

4.       Unnecessary repetition has been removed as requested, especially by Reviewer 2. For example, the conclusion section was shortened and made more general.   

Round  2

Reviewer 3 Report

Tone is much improved.

Arabic transliteration still has errors. Addition of Arabic script a plus.

Author Response

Transliteration of Arabic data has been checked thoroughly for long vowels and shaddas.